# Microbial Community Analyses Associated with Nine Varieties of Wine Grape Carposphere Based on High-Throughput Sequencing

**DOI:** 10.3390/microorganisms7120668

**Published:** 2019-12-09

**Authors:** Shiwei Zhang, Xi Chen, Qiding Zhong, Xuliang Zhuang, Zhihui Bai

**Affiliations:** 1Research Centre for Eco-Environmental Sciences, Chinese Academy of Sciences, Beijing 100085, China; swzhang@rcees.ac.cn (S.Z.); chenxi820531@126.com (X.C.); xlzhuang@rcees.ac.cn (X.Z.); 2College of Resources and Environment, University of Chinese Academy of Sciences, Beijing 100049, China; 3China National Research Institute of Food & Fermentation Industries, Beijing 100027, China; zhongqiding@163.com

**Keywords:** wine grape carposphere, varieties, community structure, illumina high-throughput sequencing

## Abstract

Understanding the composition of microbials on the grape carposphere may provide direct guidance for the wine brewing. In this work, 16S rRNA and ITS (Internal Transcribed Spacer) fungal amplicon sequencing were performed to investigate the differences of epiphytic microbial communities inhabiting different varieties of wine grape berries. The results showed that the dominated phylum of different wine grape carpospheres were Proteobacteria, Actinomycetes, Firmicutes, Gemmatimonadete, and Bacteroidetes. The dominant bacterial genera of different wine grape varieties were *Pseudomonas, Arthrobacter*, *Bacillus*, *Pantoea*, *Planomicrobium*, *Massilia*, *Curtobacterium*, *Corynebacterium*, *Cellulomonas*, *Sphingomonas*, and *Microvirga*. The fungal communities of the grapes were dominated by Ascomycota for all nine wine varieties. The dominant fungal genera on grape carposphere were *Alternaria*, *Cladosporium*, unclassified *Capnodiales, Phoma*, *Rhodotorula*, *Cryptococcus, Aureobasidium*, and *Epicoccum*. Community structure exerts a significant impact on bacterial Bray-Curtis dissimilarity on six red grapes and also a significant bacterial impact on three white grapes. Community structure exerts a significant impact on fungal Bray-Curtis dissimilarity on six red grapes but weak or no fungal impact on three white grapes. The results revealed that grape variety plays a significant role in shaping bacterial and fungal community, varieties can be distinguished based on the abundance of several key bacterial and fungal taxa.

## 1. Introduction

The microbial flora and fauna that coexist with the plants may be one of the key factors that influence the grapes size, color, flavor, yield, and so forth. There are some reports found that the microbiota presented on the carposphere may also devote to wine on the process of wine making [1]. Recently, more attention has been paid to the potential biotechnological applications of these microorganisms.

Until now, grapevine-associated microbial communities have broadly been studied with respect to pathogens [2] and the grapevine endophytic bacteria [3]. As for the carposphere, most studies have focused on the oenological interest, especially acetic and lactic acid bacteria and yeast communities in grape fruit microflora [4,5]. However, little is known about the microbial communities among varieties of wine grape carposphere. Applying the traditional method of T-RFLP, Martins et al. [3] revealed differences in the size and structure of the populations in the Merlot grape berries and other parts. Using high-throughput sequencing analysis, Portillo et al. [6] investigated the bacterial diversity on the berries of Grenache and Carignan grape varieties, indicated that heterogeneity of the microflora by grape cultivars. Zhang et al. revealed the microbial communities on Kyoho carposphere, which can be used as wine grapes and table grapes, elucidated the origin of the microbial communities on grape berries [7]. It was also reported that a large diversity of Chardonnay and Cabernet Sauvignon microbial diversity dynamic change from grape crush to wine [8].

In this study, high-throughput sequencing analysis was performed to characterize the microbial communities including bacteria and fungi on grapevine carposphere with six red wine grapes Pinot nior (Pin), Gem (Gem), Cabernet Sauvignon (Cab), Zinfandel (Zin), Syrah (Syr), Merlot (Mer) and three white wine grapes Chardonnay (Cha), Riesling (Rie), Longan (Lon). The structure and diversity of these communities were examined to evaluate the composition and abundance of bacterial as well as fungi communities.

## 2. Materials and Methods

### 2.1. Site Description

This study was performed in a large-scale wine vineyard in Huailai City, Hebei Province (40°4′–40°35′N, 115°16′–115°58′E). The location belongs to the temperate continental monsoon climate. The sunshine rate is 68%, with an average annual temperature of 9.1 °C. It has strong solar radiation during the daytime, and large temperature difference between day and night, the maximum temperature was 29 °C, and the minimum temperature was 16 °C. The growing season here extends from May to September. The vineyard is not privately-owned or protected and did not require specific permits or involve endangered or protected species. The age of the grapevines are between 10 and 37 years old.

### 2.2. Grape Sampling

Healthy and undamaged grapes of 9 varieties of Chardonnay (Cha), Riesling (Rie), Pinot nior (Pin), Gem (Gem), Cabernet Sauvignon (Cab), Zinfandel (Zin), Longan (Lon), Syrah (Syr), Merlot (Mer) were collected using sterile scissors at veraison. In this period, grapes begin to color and enlarge, corresponding to stage 35 in the modified E-L system for identifying major and intermediate grapevine growth stages [9]. Considering the heterogeneity of the tested grapes, the grape samples were collected from at least 5 plants to form a composite sample, and five composite samples of grapes were collected for each variety of grape, all the samples were placed in sterile plastic bags, transported to the laboratory in refrigerated boxes and processed within 12 h. To prevent cross contamination, sampling tools were sterilized with 75% ethanol before sampling.

### 2.3. DNA Extraction

Thirty grams of grape berries were homogenized in 100 mL sterile 0.1M PBS solution (pH 7.0) in a rocking incubator at 180 rpm for 30 min. Sonication was then performed at a frequency of 40 kHz at 4 °C for 15 min in an ultrasonic cleaning bath (Shanghai Kudos Instrument Co.) to dislodge the microbes. The bacteria and fungi in the solution were collected with 0.22 μm filters by vacuum filtration. Genomic DNA was extracted from all the samples using the FastDNA SPIN Kit for Soil (California, CA, USA), in accordance with the manufacturer’s guidelines.

### 2.4. PCR Amplification and Sequencing

PCR amplification was performed using the bacterial 16S rRNA gene V5–V7 region primers 799F (5′-AAC MGG ATT AGA TAC CCK G-3′) [10] and 1193R (5′-ACG TCA TCC CCA CCT TCC-3′) [11]. Fungal rRNA internal transcribed spacer gene ITS1 was amplified with primers 5′-CTT GGT CAT TTA GAG GAA GTA A-3′ and 5′-GCT GCG TTC TTC ATC GAT GC–3′) [10]. The reverse primers were modified to contain a barcode [12]. A volume of 50 μL PCR reaction contained 5 μL 10× Pyrobest Buffer (Takara, Dalian, China), 1 μL DNA template, 2 μL of each primer (10 μmol/L), 4 μL dNTPs (2.5 μmol/L), 0.3 μL Pyrobest DNA Polymerase (2.5 U/μL, Takara, Dalian, China), and 35.7 μL ddH_2_O. PCR reaction procedure for bacterial 16S rRNA gene consisted of an initial 94 °C for 3 min, followed by 28 cycles of 94 °C for 45 s, 50 °C for 60 s, and 72 °C for 90 s, and a final extension of 72 °C for 10 min. PCR reaction procedure for fungal rRNA gene ITS1 consisted of an initial 95 °C for 2 min followed by 40 cycles of 95 °C for 30 s, 55 °C for 30 s, and 72 °C for 60 s and a final extension of 72 °C for 5 min. PCR products were cleaned using the UltraClean PCR Clean-up Kit (Carlsbad, CA, USA).

The cleaned PCR products were passed to Beijing Fixgene Techology Co., Ltd. and sequenced using an Illumina Hiseq instrument with a PE250 paired-end (Hiseq 2500, PE250).

### 2.5. Data Analysis

The standard operating procedure from the software package MOTHUR (version 1.36.1) was applied, which includes a de-noising step (version 1.36.1) [13]. Raw sequence reads were quality trimmed with the following criteria: (a) sequences were not allowed any ambiguous bases and maximum homopolymer length was set to 8 base pairs; (b) when aligning paired ends, a maximum of 2 mismatches were allowed; (c) sequences were checked for chimeras. Sequences were split into operational taxonomic units (OTUs) based on 0.03 cut-off value and classified using SILVA bacterial 16S rRNA gene or the UNITE fungal ITS database. Diversity within the samples (alpha diversity) was calculated using ACE, Chao 1, Shannon and Simpson indices, respectively. Diversity between samples (beta diversity) was estimated by a series of tests. Community similarities based OTU using a principal coordinates analysis (PCoA) based on Bray-Curtis distance matrices as well as PerMANOVA analysis. A linear discriminant analysis effect size (LEfSe) was applied to the OTU table (non-parametric factorial Kruskal-Wallis (KW) sum-rank test *p* < 0.05, LDA > 3.5; http://huttenhower.sph.harvard.edu/galaxy/, linear discriminant analysis (LDA)) to identify the discriminant bacterial and fungi clade [14]. A co-occurrence network analysis was performed for each microbiome associated with the rhizosphere to expound the significant relations among the OTUs [15], to build the network, we filtered out the OTUs with frequencies less than 0.05, and with the spearman correlation coefficients higher than 0.85 [16] and visualized using Gephi 0.9.2. The sequencing data have been deposited in the NCBI database (https://www.ncbi.nlm.nih.gov/) and may be accessed using the bacterial accession number SRP104559 and fungi accession number SRP151319.

The core OTUs were defined as the OTUs whose sequences accounted for more than 1%, and aligned using BLAST with the NCBI. The statistical significance of the differences was tested by one-way ANOVA followed by Tukey’s test (*p* < 0.05), performed using SPSS 21.0.

## 3. Results

### 3.1. Richness and Diversity Analysis

On average, for bacterial communities on the grape samples, at least 52,256 sequences and 85,644 at most, were obtained from different varieties (Table 1). For fungal communities, at least 143,876 sequences and 611,844 at most were obtained (Table 2). Sequences were clustered into OTUs obtained 1897 bacterial and 1313 fungal OTUs at the 0.03 distance cut-off, which is typically the species level [17].

All the rarefaction curves began to level off, suggesting that the bacterial and fungal communities were reasonably characterized with the sampling effort (Appendix A). The values of coverage for bacterial communities were larger than 97% for all the samples (Table 1), and for fungal communities, they were as high as 99% (Table 2). For the bacterial communities, Cha showed the highest richness from the results of ACE and Chao 1, while Syr showed the highest diversity from the results of Shannon and Simpson (Table 1). For the fungal communities, no matter for the richness or diversity, Lon was much higher than other varieties (Table 2).

### 3.2. Microbial Community Analysis

Comparing the bacterial communities in different varieties of wine grape samples, it was found that the main phyla and genera were present in all samples, but in different amounts (Figure 1a,b), whereas others were sample-specific. At the phylum level, the bacterial communities of the grape samples were dominated by Proteobacteria, no matter for red varieties (Gem, 57.4%; Pin, 45.4%; Syr 42.2%; Zin, 53.5%; Mer, 43.3% and Cab, 50.7%) or white varieties (Lon, 37.7%; Rie, 80.5% and Cha, 57.4%), followed by Actinomycetes, Firmicutes, Gemmatimonadete, and Bacteroidetes (Figure 1a). The dominant 11 bacterial genera of different wine grape varieties were *Pseudomonas, Arthrobacter, Bacillus, Pantoea, Planomicrobium, Massilia, Curtobacterium, Corynebacterium, Cellulomonas, Sphingomonas*, and *Microvirga* (Figure 1b). It was noted that among them, the most abundant bacterial genus was *Pseudomonas*, such as for Rie, which accounted for nearly 70% of the whole bacterial community. Sample type was found to be the major explanatory variable (14.54% explained) of microbial community structure (Figure 1c) (PCo1 7.73% and PCo2 6.81%).

Compared to that of the bacterial communities, the diversity of fungi on the grape samples was far less, whether at the phylum or at the genus level (Figure 2a,b). At the phylum level, the fungal communities of the grapes were dominated by Ascomycota for all nine wine varieties (Gem, 91.37%; Pin, 92.9%; Syr, 90.5%; Zin, 99.2%; Mer, 98.3%; Cab, 98.9%; Lon, 88.6%; Rie, 92.4%, and Cha, 93.9%), followed by Basidiomycota (Figure 2a). The dominant 8 fungal genera on grape carposphere were *Alternaria*, *Cladosporium*, unclassified *Capnodiales, Phoma, Rhodotorula*, *Cryptococcus, Aureobasidium*, and *Epicoccum* (Figure 2b). Among them, the abundance of *Cladosporium* was significantly different among varieties (Gem, 24.8%; Pin, 24.5%; Syr, 13.1%; Zin, 46.4%; Mer, 32.5%; Cab 32.3%; Lon, 8.3%; Rie, 23.2%, and Cha, 19.7%) (Figure 2b). Sample type was found to be the major explanatory variable (41.44% explained) of microbial community structure (Figure 2c) (PCo1 30.85% and PCo2 10.59%).

### 3.3. Analysis of Core OTUs and Most Similar Species

The core OTUs were defined as the OTUs whose sequences accounted for more than 1% of all the sequences on the grape samples of all the varieties. Twenty-one bacterial core OTUs were obtained here, who constituted more than 60% of the total sequences (Figure 3, Appendix A). Taken these OTUs blasted in the NCBI, it was found that among all the bacterial core OTUs, the relative abundance of two taxa *Pseudomonas* sp. (OTU1, OTU10) and *Arthrobacter* sp. (OTU4, OTU7, OTU9) were found to be higher than others. The relative abundance of *Pseudomonas* sp. (OTU1, OTU10) was between 15.9% (Lon) and 58.8% (Rie), and *Arthrobacter* sp. (OTU4, OTU7, OTU9) between 5.1% (Gem) and 10.7% (Lon).

There were twenty-eight fungal core OTUs found here (Figure 4, Appendix A). These OTUs constituted most of the sequences, which was as high as 99.3% for Cab. It was found that among all the fungal core OTUs, the relative abundance of the two taxa *Alternaria* sp. (OTU1, OTU4) and *Cladosporium* sp. (OTU2, OTU5, OTU9) were found to be higher than others. The relative abundance of *Alternaria* sp. (OTU1, OTU4) was between 42.7% (Lon) and 68.3% (Cha), and *Cladosporium* sp. (OTU4, OTU7, OTU9) between 21.6% (Cha) and 36.0% (Cab).

### 3.4. Analysis the Differences among Different Varieties of Wine Grape Carposphere

Comparison of the bacterial OTUs shared among the different sample types showed that the unique OTUs of Mer, Syr, Zin, Cab, Gem, Pin, Cha, Rie and Lon were 1, 46, 2, 8, 4, 10, 3, 26 and 73, respectively. The shared OTUs of the nine wine grape varieties were 30, which took up 1.6% of the total OTUs (Figure 5).

The co-occurrence network presents the co-occurrence relationship between all samples and species. The node in the network represents the sample node and the species node, and the connection between the sample node and the species node represents that the species is included in the sample. As to the analysis of bacterial co-occurrence network, there were 1062 nodes and 4950 edges. 60.5% of the OTUs found in Mer and 0.2% of the OTUs found in Rie (Figure 6). The results showed that most of the OTUs were contributed by Mer, while the least was contributed by Rie.

Comparison of the fungal OTUs shared among the different sample types showed that the unique OTUs of Mer, Syr, Zin, Cab, Gem, Pin, Cha, Rie, and Lon were 4, 3, 6, 8, 4, 9, 13, 5, and 10, respectively. The shared OTUs of the nine wine grape varieties were 754, which took up 57.4% of the total OTUs (Figure 7).

As to the analysis of fungal co-occurrence network, there were 73 nodes and 159 edges. 21.9% of the OTUs found in Mer and 4.7% of the OTUs found in Pin and Gem, respectively (Figure 8). The network showed that the different varieties contributions microorganisms, with Mer having the most and Pin as well as Gem having the least. It also showed that from Figure 6 and Figure 8, most of the co-occurrence microorganisms were bacteria rather than fungi.

### 3.5. Differences in Bacterial and Fungal Diversity among Varieties

The linear discriminant analysis effect size (LEfSe) was able to search for statistically significant biomarker among groups, i.e., species with significant differences among groups. LEfSe statistical results include three parts, namely, the linear discriminant analysis (LDA) value distribution histogram, evolutionary branch graph (phylogenetic distribution), and biomarker abundance comparison chart in different groups. The LEfSe was used to identify discriminative bacteria taxon among different grape varieties. The LEfSe of all species showed 75 bacterial taxa with significant differences. Nine groups of bacterial were significantly enriched in Mer, *Clostridia* (class), *Bacteroidetes* (phylum), *Peptostreptococcaceae* (family), *Sphingomonas* (genus), *Corynebacteriaceae* (family), *Paeniclostridium* (genus), *Clostridiales* (order), *Romboutsia* (genus), and *Corynebacteriales* (order); three groups of bacterial were significantly enriched in Zin, *Microbacteriaceae* (family), *Curtobacterium* (genus), and *Acidimicrobiales* (order); 30 groups of bacterial were significantly enriched in Syr, *Gemmatimonadetes* (class), *Nocardioides* (genus), *Nocardioidaceae* (family), *Thermoactinomycetaceae* (family), *Erysipelotrichales* (order), *Deltaproteobacteria* (class), *Rhizobiales* (order), *Streptomycetaceae* (family), *Sphingomonadales* (order), *Propionibacteriaceae* (family), *Intrasporangiaceae* (family), *Methylobacteriaceae* (family), *Erysipelotrichia* (class), *Escherichia_Shigella* (genus), *Propionibacteriales* (order), *Alphaproteobacteria* (class), *Planomicrobium* (genus), *Ornithinimicrobium* (genus), *Gemmatimonadetes* (phylum), *Streptomyces* (genus), *Erysipelotrichaceae* (family), *Streptomycetales* (order), *Turicibacter* (genus), *Sphingomonadaceae* (family), *Micromonosporaceae* (family), *Planococcaceae* (family), *Kocuria* (genus), *Micromonosporales* (order), *Novosphingobium* (genus), and *Rhodospirillales* (order); seven groups of bacterial were significantly enriched in Rie, *Pseudomonadaceae* (family), *Proteobacteria* (phylum), *Pseudomonas* (genus), *Gammaproteobacteria* (class), *Pseudomonadales* (order), *Gammaproteobacteria* (class), and *Pantoea* (genus); 18 groups of bacterial were significantly enriched in Lon, *Frankiales* (order), *Comamonadaceae* (family), *Geodermatophilus* (genus), *Frankiales* (order), *Enterococcus* (genus), *Comamonadaceae* (family), *Micrococcales* (order), *Cellulomonas* (genus), *Actinobacteria* (phylum), *Arthrobacter* (genus), *Cellulomonadaceae* (family), *Actinobacteria* (class), *Lactobacillales* (order), *Geodermatophilaceae* (family), *Enterococcaceae* (family), *Micrococcaceae* (family), *Pseudarthrobacter* (genus), and *Blastococcus* (genus); four groups of bacteria were significantly enriched in Pin, *Betaproteobacteria* (class), *Oxalobacteraceae* (family), *Burkholderiales* (order), and *Massilia* (genus); four groups of bacterial were significantly enriched in Gem, *Sphingobium* (genus), *Xanthomonadales* (order), *Exiguobacterium* (genus), *Bacillales* (order) (Figure 9a,b). It was shown that most of the wine grapes could be clearly distinguished by the demonstrative microorganism at the genus level.

Similarly, The LEfSe was used to identify discriminative fungi taxon among different grape varieties. The LEfSe of all species showed 35 fungal taxa with significant differences. two groups of fungal were significantly enriched in Lon, *Capnodiales* (order) and *Davidiellaceae* (family); two groups of fungal were significantly enriched in Cab, *Dothideomycetes* (class) and *Ascomycota* (phylum); nine groups of fungal were significantly enriched in Syr, *Guehomyces* (genus), *Gibberella* (genus), *Zygomycota* (phylum), *Mortierellales* (order), *Cystofilobasidiales* (order), *Mortierellaceae* (family), *Mortierella* (genus), *Cystofilobasidiaceae* (family), and *Zygomycota* (phylum); 19 groups of fungal were significantly enriched in Lon, *Hypocreales* (order), *Xylariales* (order), *Preussia* (genus), *Tremellales* (order), *Basidiomycota* (phylum), *Hypocreales* (order), *Monographella* (genus), *Xylariales* (order), *Sordariales* (order), *Fusarium* (genus), *Nectriaceae* (family), *Sordariomycetes* (class), *Sordariales* (order), *Tremellomycetes* (class), *Tremellales* (order), *Sporormiaceae* (family), *Hypocreales* (order), *Sordariales* (order), and *Cryptococcus* (genus); two groups of fungal were significantly enriched in Pin, *Cladosporium* (genus) and *Sporormiaceae* (family) (Figure 10a,b). It was shown that the wine grapes could be clearly distinguished by the demonstrative microorganism at least at the phylum and class level with the variety of Cab and Syr, while some varieties need to be distinguished at the genus level, such as Lon and Pin.

In order to elucidate the differences in bacterial and fungal compositions among different grape varieties, the Bray-Curtis dissimilarity was adopted to study the most common wine grapes in China, Cab, Cha, Zin, Syr, Gem, Rie, Lon, Mer, and Pin separately, which are also common in the world, and were analyzed independently to dissect intravarietal biogeographical relationships. The patterns in grape carposphere microbiota suggest a genetic component to host-microbial interactions on the grape surface. We calculated taxonomic metric (Bray-Curtis dissimilarities calculated from OTU), tests showed that community structure varied among different varieties, exerting a significant impact of bacterial Bray-Curtis on six red grape (R_adonis_ = 0.460, *p* < 0.01, R_anosim_ = 0.529, *p* < 0.01), there was also a significant bacterial impact on three white grapes (R_adonis_ = 0.492, *p* < 0.01, R_anosim_ = 0.549, *p* < 0.01). For the fungal Bray-Curtis, tests showed that community structure varied among different varieties, exerting a significant impact of fungal Bray-Curtis on six red grape (R_adonis_ = 0.548, *p* < 0.01, R_anosim_ = 0.398, *p* < 0.01), there was weak or no fungal impact on three white grapes (R_adonis_ = 0.249, *p* > 0.01, R_anosim_ = 0.207, *p* > 0.01) (Table 3). These results revealed that grape variety plays a significant role in shaping bacterial and fungal community, varieties can be distinguished based on the abundance of several key bacterial and fungal taxa.

## 4. Discussion

There are some researches on the microbial populations of wine grapes, while little is known about the typical different varieties of wine grape among epiphytic microorganisms of grapevine. In this work, high-throughput sequencing analysis was performed to investigate the differences of epiphytic microbial communities inhabiting different varieties of wine grape berries. This approach revealed both similarities and differences in population diversity and composition among different varieties of samples.

### 4.1. The Richness and Diversity in the Wine Grape Berries of Different Varieties

Marasco et al. [18] revealed the richness and diversity of ungrafted grapevine and grafted grapevine are significantly influenced by the species. In this study, the results of bacterial alpha diversity showed that for the microbial communities, the richness (Table 1) on Cha berry samples was higher than that on other grape samples, followed by that on Rie, the Cab and Syr were lowest. The results of fungal alpha diversity showed that for the microbial communities, the richness and diversity (Table 2) on Lon grape was higher than that on other berry samples, and the Cha was lowest.

Thirty OTUs were detected in all of the bacterial samples, and 21 OTUs (97% similarity) were found in 60% of the samples. These “core” microbiota, comprising OTUs associated with *Pseudomonas* sp. and *Arthrobacter* sp. were more abundant in Lon, Cha, and Cab. Every sample type also had unique OTUs (Mer 1, Syr 46, Zin 2, Cab 8, Gem 4, Pin 10, Cha 3, Rie 26, and Lon 73), *Pseudomonas* were found more frequently on grape berries and 30 unique OTUs were also found in Mer [19].

Seven hundred and fifty four OTUs were detected in all of the fungal samples, and 28 OTUs (97% similarity) were found in 60% of the samples. These “core” microbiota, comprising OTUs associated with *Alternaria* sp., *Cladosporium* sp., and *Cladosporium* sp. were more abundant in Lon, Cha, and Cab. Every sample type also had unique OTUs (Mer 4, Syr 3, Zin 6, Cab 8, Gem 4, Pin 9, Cha 13, Rie 5, and Lon 10). *Alternaria* produces alternariol, alternariol monomethyl, altenuene, altertoxin, and tenuazonic acid, these metabolites exhibit some degree of toxicity to mammalian and bacterial cells as well as to higher plants [20]. *Fusarium* sp. are involved in wine making and can produce pectinase, raising juice yields during the process of wine making [21,22]. In our study, all of these two potential functional microorganisms were found in the nine varieties of wine grapes.

### 4.2. Bacterial and Fungal Taxa Distribution in the Grape Carposphere Are Significantly Different with the Varieties

Comparison of the microbial communities in nine wine grape carpospheres showed that the main phyla and genera were present in all varieties, but in different amounts. Of all varieties, the relative abundance of *Pseudomonas, Arthobacter*, and *Bacillus* were the three highest genus, the presence of *Pseudomonas* in Riesling was as high as 66.0%. *Pseudomonas* can survival in low-nutrient conditions [1,3] and certain *Pseudomonas* were found more frequently on grape berries [23]. Why is it more abundant in Riesling. There may be two reasons as follows: Firstly, the harvest season of Riesling is commonly later than other varieties, it is even harvested until November. Secondly, Riesling can grow well in rocks and foggy conditions, which we called an extreme-environment. *Bacillus* may play a key role during alcohol fermentation [24]. Leveau [23] found that *Massilia* was the dominant genus on Chardonnay, in our study, it was found that the *Massilia* in Syr was 8.62%, while in Cha it was just 0.50%. *Sphingomonas* can survive the wine fermentation process [25] and yet its impacts on wine organoleptic properties remain unknown.

At the level of fungi genus, Zhang et al. [7] found *Alternaria* was enriched in the Kyoho grape. In our study, it was also found that *Alternaria* was enriched in all of the samples, whilst it was different with the results of Martins [26]. The sampling time of Martins was in the ripening period of autumn, while the sampling time of Zhang et al. [7] and our study was in July, the time of grape blooming. At this time, the period of color conversion of grape berries, berries are no longer swelling during this period and the chlorophyll content in the peel is decomposing, basically turned into red and dark blue. The white grape varieties are transparent, and the sugar content of berries increases linearly. This is a turning point for most grape varieties with reduced acid content.

### 4.3. The Grape Variety Influences the Microorganisms on the Grape Berries and Further Influences the Quality of Wine Grape

Revealing cultivar is suspected to decrease pressures under complex environments. For instance, the acetate production by *Gluconobacter* [27], malate conversion, off-flavor production by lactic acid bacteria [1,28], or the desirable enhancement of the sensory profile by *C.zemplinina* [29] all cover the wine-quality. Given the relationship among these taxa and specific grape varieties, the winemaker can customized fermentation management strategies that improve product outcomes. No matter if it is red wine grapes or white wine grapes, *Pseudomonas*, *Arthrobacter*, and *Bacillus* were the most abundant genus associated with wine grapes of all genus that are typical of the aerial part of plants [30,31]. The origin of the microbes in wine ferments is still poorly understood, they are commonly assumed to come from the grapes themselves [32]. The present study found that most of the grapes have more species, a plausible explanation for this might be that all of the grapes grow in the same field.

Comparison of the communities associated with the red grapes and white grapes revealed both ubiquitous and cultivar specific groups. *Alternaria* (45.9–68.4%) in red wine grapes, (30.9–68.4%), and *Cladosporium* interestings *Alternaria* is the main disease of fungi, it can produce mycotoxin which lead to carcinngenic [33]. We found Riesling *Alternaria* and *Pseudomonas* both were higher, it may explained that *Pseudomonas* influence of fungal metabolism on bacterial physiology and survival [34]. In addition, we found differences in beta-diversity of the red wine grapes and white wine grapes, which reveal differences in the composition of microbes associated with these grapes, these differences in composition might be due to the fact that different cultivars of wine grapes have different sizes, shapes, and growth characteristics. Take the Chardonnay for example, it has thin skins and dense growth habits may result in crowding and berry breakage, while Gem has thick skins and sparse berries. The good ventilation may lead to the lower fungal diseases (such as alternaria). For both bacterial communities (R_ANOSIM_ = 0.529, *p* < 0.01) and fungi communities (R_ANOSIM_ = 0.549, *p* < 0.01), varietal variation is most powerful within a limited region (Table 3). These varietal-microbial patterns appear stable in different years [35]. Meanwhile, varietal patterns in grape-surface microbiota showed a genetic component to host-microbial interactions on the grape surface.

## 5. Conclusions

The present analysis, based on high-throughput sequencing technology, revealed the microbial communities on the carposphere of different wine grapes varieties. The results showed that Riesling has the richest microbial communities among these wine grapes. The dominant phyla of bacteria were Proteobacteria, Actinobacteria, Firmicutes, Gemmatimonadete, and Bacteroidetes and the genera were *Pseudomonas, Arthrobacter, Bacillus, Pantoea, Planomicrobium, Massilia, Curtobacterium, Corynebacterium, Cellulomonas, Sphingomonas*, and *Microvirga*. The dominant phyla of the fungi were Ascomycota, Basidiomycota, Zygomycota, and the genera were *Alternaria*, *Cladosporium*, unclassified *Capnodiales*, *Phoma*, *Rhodotorula*, *Cryptococcus*, *Aureobasidium*, and *Epicoccum.* The most similar species of the bacterial core OTUs were *Arthrobacter* sp., *Pseudomonas* sp., *Cladosporium* sp., *Phoma* sp., and *Alternaria* sp. The most similar species of the fungal core OTUs were *Alternaria* sp. and *Cladosporium* sp. From the results, it is inferred that the microorganisms existed on the carposphere of the wine grapes may be beneficial or harmful to the health of the grape vine, the quality of the grape berries, and the process of wine brewing.

## Figures and Tables

**Figure 1 microorganisms-07-00668-f001:**
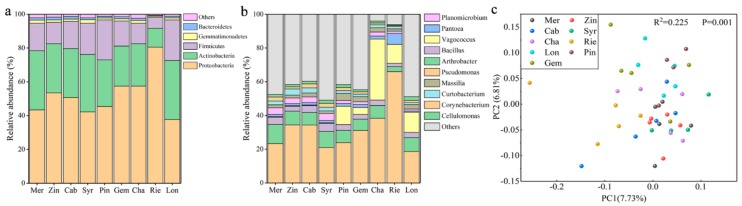
Comparison of the bacterial communities among the grape samples of different varieties at the phylum and genus level, (**a**) at the phylum level; (**b**) at the genus level. (**c**) Principal coordinate analysis (PCoA) among different varieties of wine grape carposphere based on bacterial unweighted UniFrac distances at the genus level.

**Figure 2 microorganisms-07-00668-f002:**
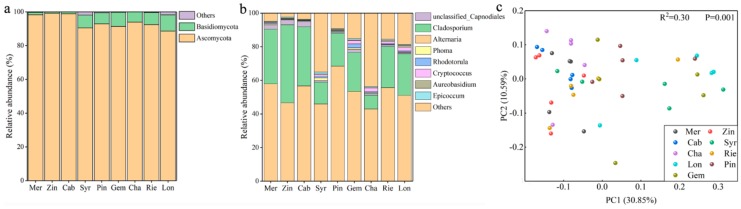
Comparison of the fungal communities among the grape samples of different varieties at the phylum and genus level. (**a**) At the phylum level; (**b**) at the genus level; (**c**) principal coordinate analysis (PCoA) among different varieties of wine grape carposphere based on fungal unweighted UniFrac distances at the genus level.

**Figure 3 microorganisms-07-00668-f003:**
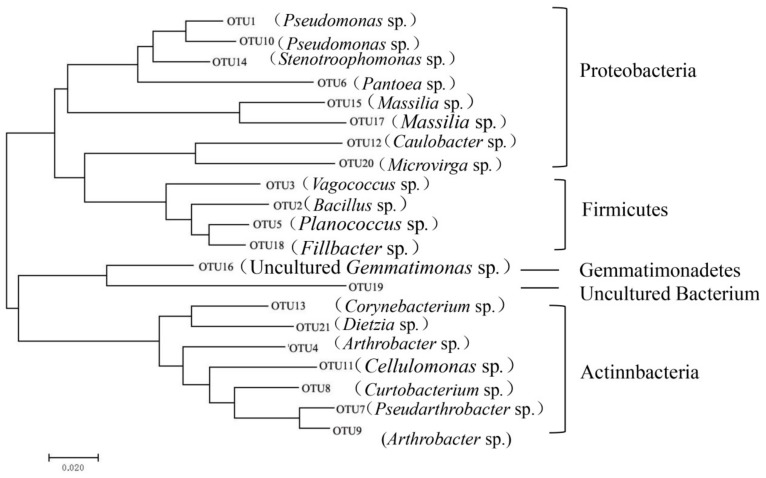
Phylogenetic tree of core bacterial communities in nine grape carposphere.

**Figure 4 microorganisms-07-00668-f004:**
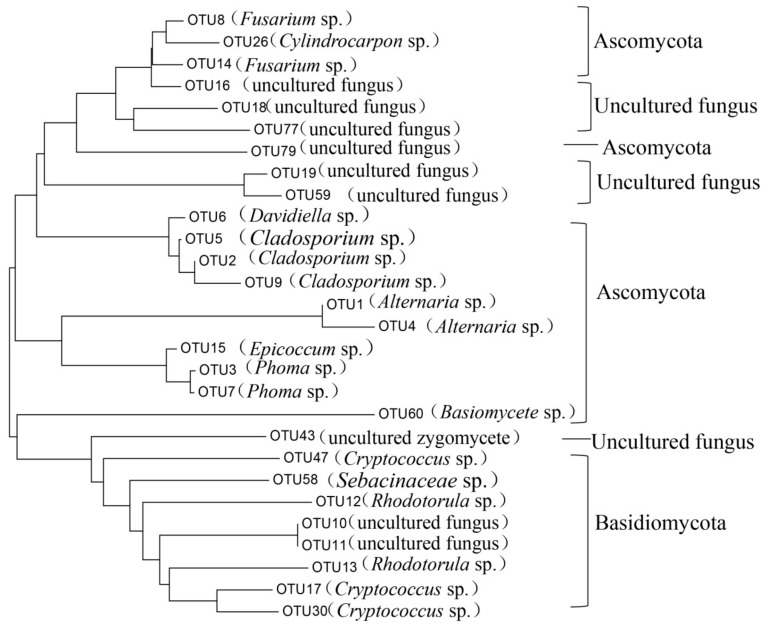
Phylogenetic tree of core fungal communities in nine grape carposphere.

**Figure 5 microorganisms-07-00668-f005:**
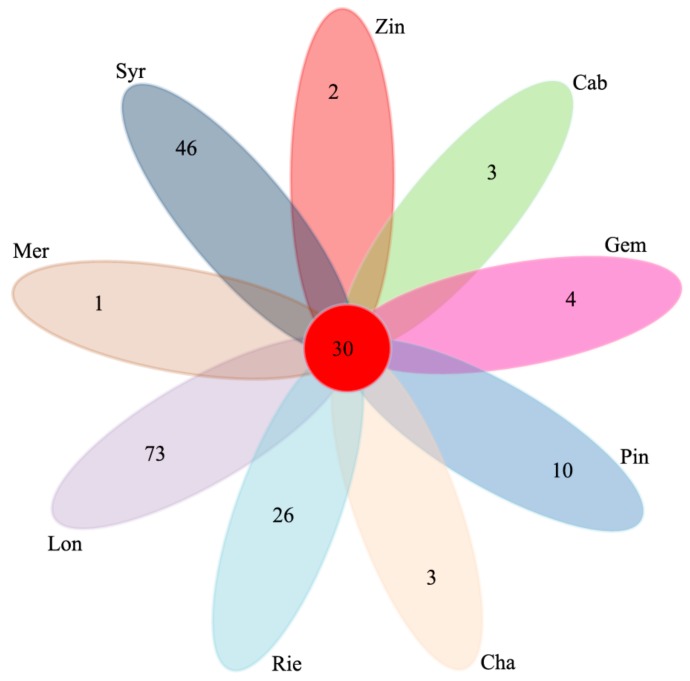
Venn diagram of operational taxonomic units (OTUs) at cut-off of 0.03 for the bacterial communities in the nine wine grape carposphere.

**Figure 6 microorganisms-07-00668-f006:**
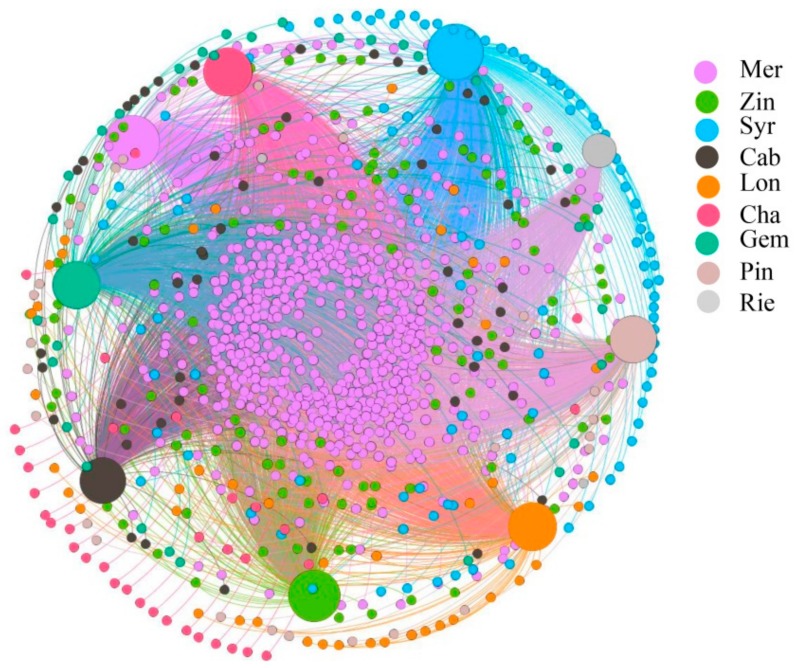
Co-occurrence network of bacterial OTU sharing between samples. Samples are represented as large circles with sample type designated by color. Edges connect sample nodes to OTU nodes detected in that sample and are also colored by sample type.

**Figure 7 microorganisms-07-00668-f007:**
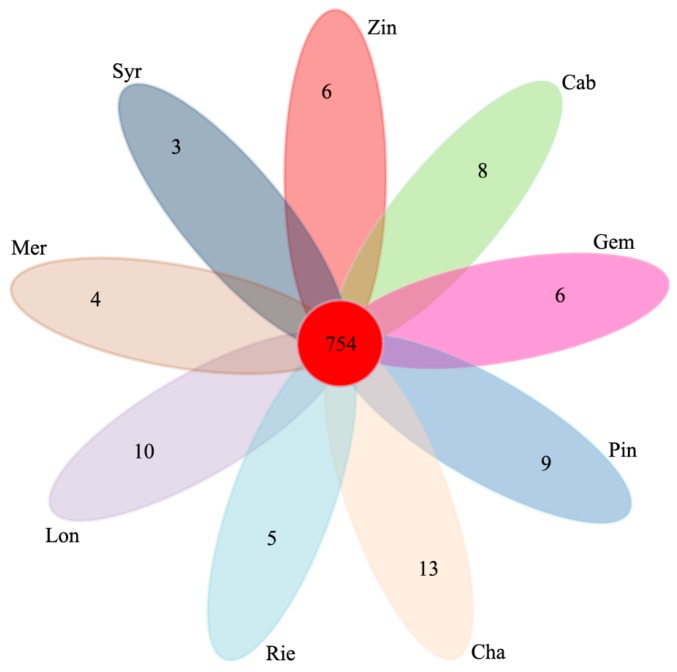
Venn diagram of OTUs at cut-off of 0.03 for the fungal communities in the nine wine grape carposphere.

**Figure 8 microorganisms-07-00668-f008:**
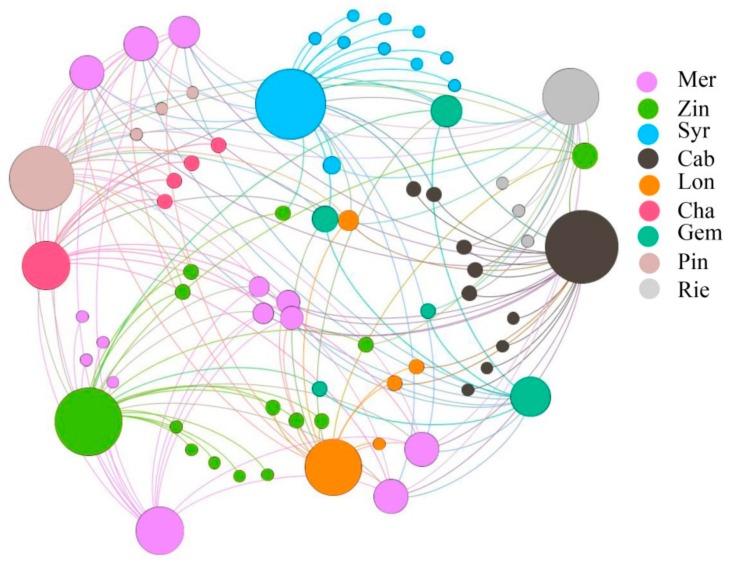
Co-occurrence network of fungal OTU sharing between samples. Samples are represented as large circles with sample type designated by color. Edges connect sample nodes to OTU nodes detected in that sample and are also colored by sample type.

**Figure 9 microorganisms-07-00668-f009:**
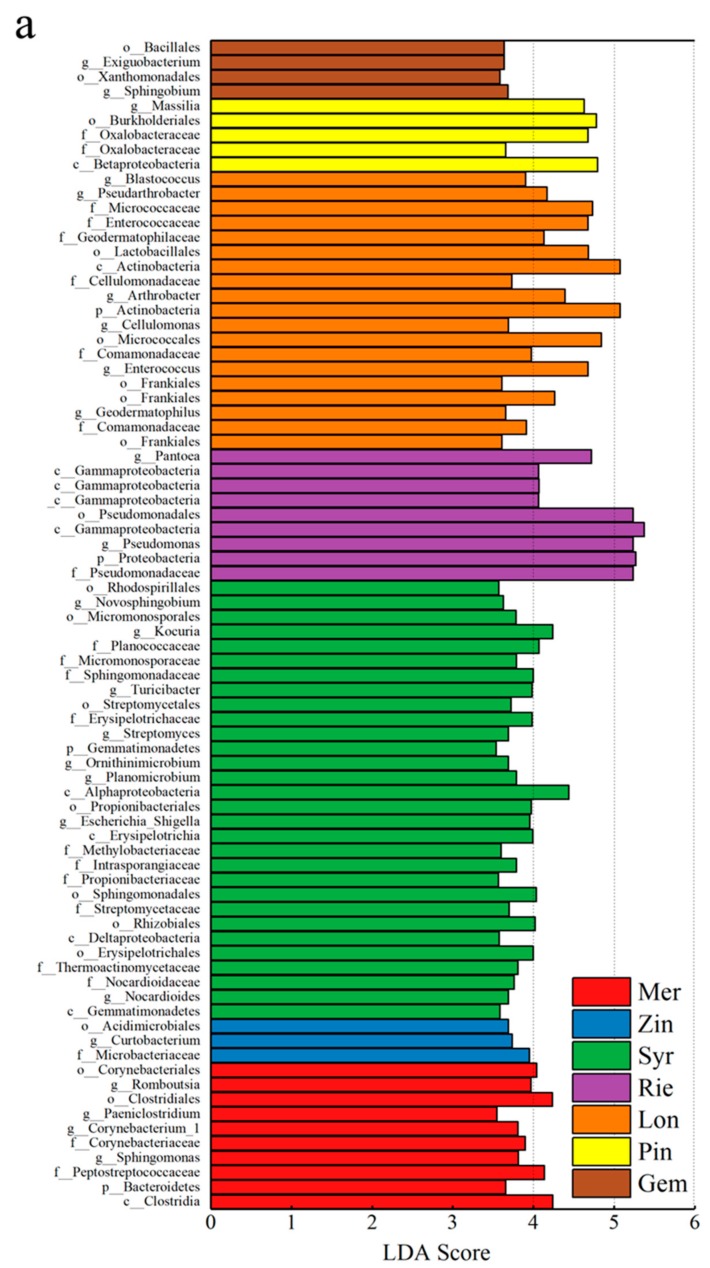
Cladograms indicating the polygenetic distribution of bacterial lineages in the carposphere of nine grape varieties as determined by linear discriminant analysis (LDA) effect size (LEfSe). (**a**), the linear discriminant analysis (LDA) value distribution histogram; (**b**), The bar charts report the taxonomic representation of statistically and biologically as determined by LEfSe analysis.

**Figure 10 microorganisms-07-00668-f010:**
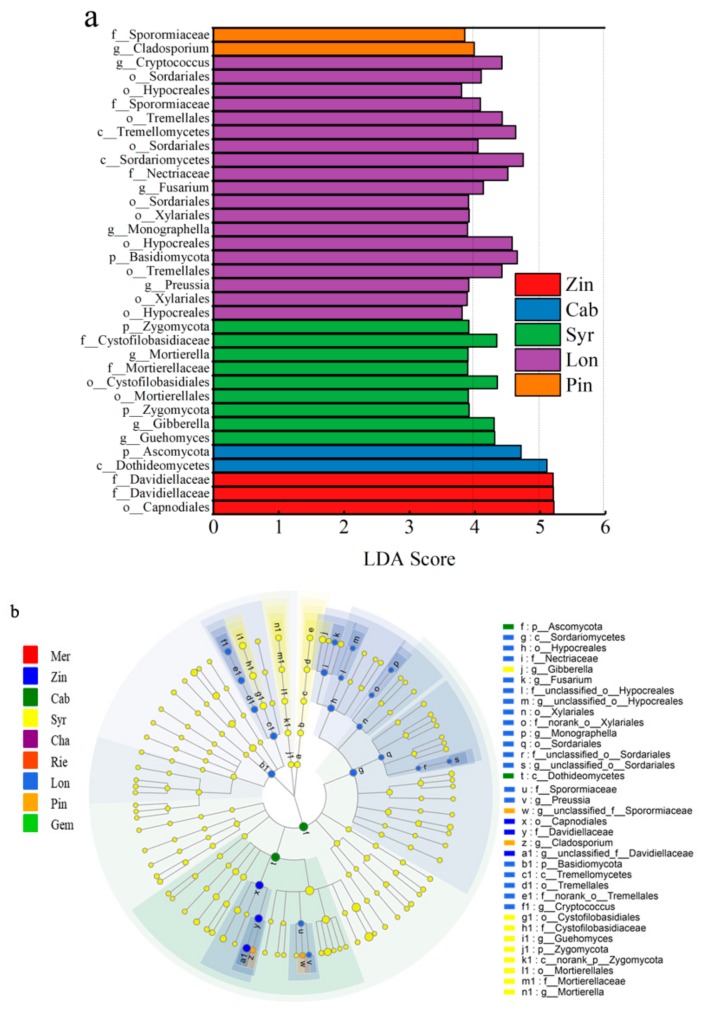
Cladograms indicating the polygenetic distribution of bacterial lineages in the carposphere of nine grape varieties as determined by linear discriminant analysis (LDA) effect size (LEfSe). (**a**), the linear discriminant analysis (LDA) value distribution histogram; (**b**), The bar charts report the taxonomic representation of statistically and biologically as determined by LEfSe analysis.

**Table 1 microorganisms-07-00668-t001:** Richness and diversity indices of bacterial communities of the grape samples of different varieties.

Variety	Nseqs (×10^4^)	Coverage (×10^−1^)	ACE (×10^3^)	Chao1 (×10^3^)	Shannon	Simpson
Cha	7.86	9.79 ± 0.0a	1.31 ± 0.16a	1.28 ± 0.16ac	4.04 ± 0.78a	0.16 ± 0.10b
Rie	8.56	9.81 ± 0.0a	1.21 ± 0.24a	1.11 ± 0.15ac	2.36 ± 0.13a	0.41 ± 0.03a
Lon	6.95	9.75 ± 0.1a	1.18 ± 0.076a	1.13 ± 0.11ac	4.29 ± 4.29a	0.10 ± 0.10a
Mer	5.31	9.73 ± 0.1a	1.18 ± 0.14a	112 ± 0.083ac	4.59 ± 0.23abc	0.06 ± 0.01a
Zin	7.12	9.85 ± 0.0a	1.24 ± 0.15a	1.15 ± 0.12ac	4.17 ± 0.33a	0.12 ± 0.05a
Cab	6.70	9.79 ± 0.0a	1.15 ± 0.090a	1.12 ± 0.093ac	4.21 ± 0.61a	0.12 ± 0.08a
Syr	6.14	9.81 ± 0.0a	1.16 ± 0.084a	1.18 ± 0.093a	4.82 ± 0.14a	0.05 ± 0.01a
Pin	5.23	9.77 ± 0.0a	1.17 ± 0.066a	1.15 ± 0.058c	4.41 ± 0.21bc	0.08 ± 0.02a
Gem	8.22	9.83 ± 0.1a	1.20 ± 0.12a	1.11 ± 0.069ac	4.09 ± 0.61a	0.13 ± 0.09a

**Table 2 microorganisms-07-00668-t002:** Richness and diversity indices of fungal communities of the grape samples of different varieties.

Variety	Nseqs (×10^5^)	Coverage (×10^−1^)	ACE (×10^3^)	Chao1 (×10^3^)	Shannon	Simpson
Cha	3.51	9.99 ± 0.0a	0.27 ± 0.043ab	20.21 ± 0.030a	1.19 ± 0.23d	0.49 ± 0.10a
Rie	4.61	9.99 ± 0.0a	0.46 ± 0.27ab	0.39 ± 0.30a	1.91 ± 0.44ade	0.31 ± 0.66ab
Lon	6.12	9.98 ± 0.0a	0.78 ± 0.26a	0.78 ± 0.28a	2.74 ± 0.64ae	0.18 ± 0.06a
Mer	3.15	9.99 ± 0.0a	0.27 ± 0.092ab	0.22 ± 0.057a	1.34 ± 0.21d	0.41 ± 0.07b
Zin	2.27	9.99 ± 0.00a	0.32 ± 0.076ab	0.25 ± 0.034a	1.33 ± 0.00c	0.38 ± 0.00ab
Cab	3.46	9.99 ± 0.0a	0.32 ± 0.031b	0.23 ± 0.028a	1.31 ± 0.10d	0.40 ± 0.03a
Syr	3.96	9.98 ± 0.0a	0.63 ± 0.27ab	0.59 ± 0.28a	2.26 ± 0.98bef	0.32 ± 0.18ab
Pin	2.55	9.97 ± 0.0a	0.64 ± 0.081ab	0.56 ± 0.13a	1.94 ± 0.46d	0.33 ± 0.07ab
Gem	1.44	9.98 ± 0.0a	0.49 ± 0.029ab	0.49 ± 0.015a	2.20 ± 0.36abd	0.22 ± 0.09ab

**Table 3 microorganisms-07-00668-t003:** Adonis and Anosim of category effects on microbial diversity patterns.

		Bacterial Bray-Curtis				Fungal Bray-Curtis			
		ADONIS		ANOSIM		ADONIS		ANOSIM	
Group	Factor	*R* ^2^	*p*	*R*	*p*	*R* ^2^	*p*	*R*	*p*
Red-Six	Variety	0.460	0.001	0.529	0.001	0.548	0.001	0.398	0.001
White-Three	Variety	0.492	0.001	0.549	0.001	0.249	0.059	0.207	0.020

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
