# Peer review of "Microbial Community Analyses Associated with Nine Varieties of Wine Grape Carposphere Based on High-Throughput Sequencing"

_microorganisms, 2019, doi:10.3390/microorganisms7120668_

Round 1
Reviewer 1 Report
This paper uses 16S rRNA gene and ITS to identify dominant bacterial and fungal communities on different grapes. In this aspect, the work is good. Unfortunately, interpretations are not made beyond this and therefore results in a publication of little usefulness and interest. Additionally, it was very difficult to read and comprehend the writing. English is a very difficult language and I want to commend the authors for trying to publish their work in a non-native language. Please go through and make sure the topic sentence describes what the paragraph will be about. Listing numbers/values is difficult and if you want to avoid this, please further describe the top taxa and focus on those while keeping the data for tables. Other suggestion would be to review the papers you cited and try to use similar words. For example, I recommend saying type of wine instead of variety. Moving to the figures, these are nicely displayed, but the ordination plots need a better coloring scheme. Similar or different shades of the same color make interpretations of the results impossible.
I hope the authors focus on cleaning up the writing (especially sentence structure and flow) and provide more interpretations of the results, and submit their work again.
Abstract
Please state that the microbes were surveyed with 16S rRNA gene and ITS analyses.
Lines 22–23 “Community structure exerting a significant impact about bacterial Bray-Curtis on six red grapes and also a significant bacterial impact on three white grapes.”
Bray-Curtis should not be used on its own. Please change to Bray-Curtis dissimilarity.
Lines 23–27 “Community structure exerting a significant impact about fungal Bray-Curtis on six red grapes but weak or no fungal impact on three white grapes.”
Bray-Curtis should not be used on its own. Please change to Bray-Curtis dissimilarity.
Introduction
Lines 32–33 “The microbial flora that coexist with the plants may be one of the key factors that influence the grapes size, color, flavor, yield and so forth.”
Flora is used to describe “prokaryotes”, but fauna is used to describe eukaryotic organisms. Yeast should be flora because they are single-celled. I suggest “The microbial flora and fauna that coexist with the plants may be one of the key factors that influence the grapes size, color, flavor, yield and so forth.”
Results
Lines 54–55 “The standard operating procedure from the software package MOTHUR (version 1.36.1) was applied, which includes a de-noising step [9].”
This should be moved to the methods section.
Figure 1 is incredibly hard to understand. There are so many colors available, but either the same or similar colors were used. It appears there is a divide in wine types, but with this color scheme I cannot make any sense of it.
Discussion
Lines 238–240 “In this work, high-throughput sequencing analysis was performed to investigate the differences of epiphytic microbial communities inhabiting different varieties of wine grape berries.”
This needs to be added/incorporated into the title and the abstract.
Lines 259–260 “3.2. Bacterial and fungal taxa distribution in the grape carposphere are significantly influenced by the varieties”
Or is it the other way around? Work wasn’t done here to support this claim.
Methods
Lines 312–313 “It has strong solar radiation during the day time, and large temperature difference between day and night.”
Day time should be daytime. Also, can you provide an average or typical difference?
Lines 313-314 “The crop growth season sustained from May to September.”
Please rephrase. I suggest “The growing season here extends from May to September.”
It is very difficult to understand the different types of wine, because the only full-word use comes at the end of the paper in the methods section. Please write the types out when they are first used in the intro before using the abbreviation.
Lines 322–323 “For each variety, five sampling points were selected randomly in the vineyard and then mixed together.”
Can you show this on a map or provide approximate location (e.g., GPS coordinates)?
Lines 323–324 “After collection, all the samples were placed in sterile plastic bags, transported to the laboratory in refrigerated boxes and processed within 12 h.”
How were they collected? Were grapes picked by hand? Where stems collected…please provide more information.
Lines 325–327 “The physicial chemical characteristics of the grape wine made from 7 varieties of the grapes collected in this study were as follows:”
The table should be cited in the text and not included in the text. Also, it is “physico-chemical”, but I do not think that is how this should be phrased because it is more of the composition of the grapes. I would just cite the table at the end of the first sentence in this paragraph. Example “Healthy and undamaged grapes of 9 varieties of Chardonnay(Cha), Riesling(Rie), Pinot nior(Pin), Gem(Gem), Cabernet Sauvignon(Cab), Zinfandel(Zin), Longan (Lon), Syrah (Syr), Merlot(Mer) were collected at veraison.”
I also have three concerns with the table.
Why are Chardonnay and Merlot types left out? Why is Pinot nior and Zinfandel missing data? I would present the wines types in a consistent order throughout the paper.Lines 360–361 “Diversity between samples (beta diversity) was estimated by a series of tests, including ANOVA and Tukey’s test, which were performed using Bray-Curtis distances.”
Bray-Curtis distances does not exist. We use a Bray-Curtis dissimilarity metric and show distance with an ordination plot.
Conclusion
Lines 371–373 “The bacteria mainly distributed in eight phyla, including Proteobacteria, Actinobacteria, Firmicutes, and six genera, such as Erwinia, Pseudomonas, Bacillus Cohn, and so on.”
Please rephrase to include the names of all phyla and genera. Otherwise, it places emphasis on these taxa.
The abstract and conclusions do not match up:
Abstract - “The dominated phylum of different wine grape carposphere were Proteobacteria, Actinomycetes, Firmicutes, Gemmatimonadete and Bacteroidetes. The dominant bacterial genera of different wine grape varieties were Pseudomonas, Arthrobacter, Bacillus, Pantoea, Planomicrobium, Massilia, Curtobacterium, Corynebacterium, Cellulomonas, Sphingomonas and Microvirga. The fungal communities of the grapes were dominated by Ascomycota for all nine wine varieties. The dominant fungal genera on grape carposphere were Alternaria, Cladosporium, unclassified Capnodiales, Phoma, Rhodotorula, Cryptococcus, Aureobasidium and Epicoccum”
Conclusion – “The bacteria mainly distributed in eight phyla, including Proteobacteria, Actinobacteria, Firmicutes, and six genera, such as Erwinia, Pseudomonas, Bacillus Cohn, and so on. The dominant phyla of the fungi were Ascomycota, Basidiomycota, Zygomycota, and the genera were Alternaria, Cladosporium, Phoma, Fusarium. The most similar species of the bacterial core OTUs were Arthrobacter sp., Pseudomonas sp., and Cladosporium sp., Phoma sp., Alternaria sp”
Lines 367–368 “From the results, it is inferred that the microorganisms existed on the carposphere of the wine grapes may be beneficial or harmful on the health of the grape vine, the quality of the grape berries and the process of wine brewing.”
Analysis of the taxonomy is good, but where is the interpretation? This conclusion is not helpful and based on the relative abundances of the top taxa you could easily make assumptions as to their impact.
Author Response
This paper uses 16S rRNA gene and ITS to identify dominant bacterial and fungal communities on different grapes. In this aspect, the work is good. Unfortunately, interpretations are not made beyond this and therefore results in a publication of little usefulness and interest. Additionally, it was very difficult to read and comprehend the writing. English is a very difficult language and I want to commend the authors for trying to publish their work in a non-native language. Please go through and make sure the topic sentence describes what the paragraph will be about. Listing numbers/values is difficult and if you want to avoid this, please further describe the top taxa and focus on those while keeping the data for tables. Other suggestion would be to review the papers you cited and try to use similar words. For example, I recommend saying type of wine instead of variety. Moving to the figures, these are nicely displayed, but the ordination plots need a better coloring scheme. Similar or different shades of the same color make interpretations of the results impossible. I hope the authors focus on cleaning up the writing (especially sentence structure and flow) and provide more interpretations of the results, and submit their work again.
Abstract
Point 1: Please state that the microbes were surveyed with 16S rRNA gene and ITS analyses.
Response: Thanks for your issue. We have added the sentence “In this work, high-throughput sequencing analysis was performed to investigate the differences of epiphytic microbial communities inhabiting different varieties of wine grape berries” in the Abstract.
Point 2: Lines 22–23 “Community structure exerting a significant impact about bacterial Bray-Curtis on six red grapes and also a significant bacterial impact on three white grapes.”
Bray-Curtis should not be used on its own. Please change to Bray-Curtis dissimilarity.
Response: Thanks for your issue. The express “Bray-Curtis” have been changed to “Bray-Curtis dissimilarity”.
Point 3: Lines 23–27 “Community structure exerting a significant impact about fungal Bray-Curtis on six red grapes but weak or no fungal impact on three white grapes.”
Bray-Curtis should not be used on its own. Please change to Bray-Curtis dissimilarity.
Response: Thanks for your issues. The express “Bray-Curtis” have been changed to “Bray-Curtis dissimilarity”.
Introduction
Point 4: Lines 32–33 “The microbial flora that coexist with the plants may be one of the key factors that influence the grapes size, color, flavor, yield and so forth.”
Flora is used to describe “prokaryotes”, but fauna is used to describe eukaryotic organisms. Yeast should be flora because they are single-celled. I suggest “The microbial flora and fauna that coexist with the plants may be one of the key factors that influence the grapes size, color, flavor, yield and so forth.”
Response: Thanks for your issue. The express “The microbial flora that coexist with the plants may be one of the key factors that influence the grapes size, color, flavor, yield and so forth” have been changed to “The microbial flora and fauna that coexist with the plants may be one of the key factors that influence the grapes size, color, flavor, yield and so forth”.
Results
Point 5: Lines 54–55 “The standard operating procedure from the software package MOTHUR (version 1.36.1) was applied, which includes a de-noising step [9].”
This should be moved to the methods section.
Response: Thanks for your issue. The sentence “The standard operating procedure from the software package MOTHUR (version 1.36.1) was applied, which includes a de-noising step [9]” have been moved to the methods section in the line of 371-372: The standard operating procedure from the software package MOTHUR (version 1.36.1) was applied, which includes a de-noising step (version 1.36.1)[9].
Point 6: Figure 1 is incredibly hard to understand. There are so many colors available, but either the same or similar colors were used. It appears there is a divide in wine types, but with this color scheme I cannot make any sense of it.
Response: Thanks for your issue. We have redrew the Figure 1 and Figure 2, and added the statistical analysis at the genus level.
Figure 1. Comparison of the bacterial communities among the grape samples of different varieties at the phylum and genus level, a, at the phylum level; b, at the genus ;level. c, Principal coordinate analysis (PCoA) among different varieties of wine grape carposphere based on bacterial unweighted UniFrac distances at the genus level.
Figure 2. Comparison of the fungal communities among the grape samples of different varieties at the phylum and genus level. a, at the phylum level; b, at the genus level; c, Principal coordinate analysis (PCoA) among different varieties of wine grape carposphere based on fungal unweighted UniFrac distances at the genus level.
Discussion
Point 7: Lines 238–240 “In this work, high-throughput sequencing analysis was performed to investigate the differences of epiphytic microbial communities inhabiting different varieties of wine grape berries.”
This needs to be added/incorporated into the title and the abstract.
Response: Thanks for your issue. According to your advise, the tittle has been changed to “Microbial community analysises associated with nine varieties of wine grape carposphere based on high-throughput sequencing”, and we also added the sentence “In this work, high-throughput sequencing analysis was performed to investigate the differences of epiphytic microbial communities inhabiting different varieties of wine grape berries.”.. to the abstract.
Point 8: Lines 259–260 “3.2. Bacterial and fungal taxa distribution in the grape carposphere are significantly influenced by the varieties”
Or is it the other way around? Work wasn’t done here to support this claim.
Response: Thanks for your issue. In this section, we found that though similarity, there are some differences among varieties. So we changed the subtittle to “Bacterial and fungal taxa distribution in the grape carposphere are significantly different with the varieties”
Methods
Point 9: Lines 312–313 “It has strong solar radiation during the day time, and large temperature difference between day and night.”
Day time should be daytime. Also, can you provide an average or typical difference?
Response: Thanks for your issue. According to your advise, we have changed the word “day time” to “daytime”. And the typical difference of the place is large temperature difference between day and night, the maximum temperature 29°C, and the minimum temperature was 16 °C.
Point 10: Lines 313-314 “The crop growth season sustained from May to September.”
Please rephrase. I suggest “The growing season here extends from May to September.”
It is very difficult to understand the different types of wine, because the only full-word use comes at the end of the paper in the methods section. Please write the types out when they are first used in the intro before using the abbreviation.
Response: Thanks for your issue. According to your advise, we have changed the express “The growing season here sustained from May to September” to “The growing season here extends from May to September”. We have write the types out in the intro before using the abbreviation. “In this study, high-throughput sequencing analysis was performed to characterize the microbial communities including bacteria and fungi on grapevine carposphere with six red wine grapes Pinot nior (Pin), Gem (Gem), Cabernet Sauvignon (Cab), Zinfandel (Zin), Syrah (Syr), Merlot (Mer) and three white wine grapes Chardonnay (Cha), Riesling (Rie), Longan (Lon).”
Point 11: Lines 322–323 “For each variety, five sampling points were selected randomly in the vineyard and then mixed together.”
Can you show this on a map or provide approximate location (e.g., GPS coordinates)?
Response: Thank you for your issue. All the samples we collected were between the attitude 40°4'-40°35'N, 115°16'-115°58'E, when sampling, we chose the health grapevine randomly. Considering the heterogeneity of the tested grapes, the grape samples were collected from at least 5 plants to form a composite sample, and five composite samples of grapes were collected for each variety of grape, that is, every variety is not a single point, but a composite sample composed of multiple points. So it is unable to mark the specific location of each variety very accurately.
Point 12: Lines 323–324 “After collection, all the samples were placed in sterile plastic bags, transported to the laboratory in refrigerated boxes and processed within 12 h.”
How were they collected? Were grapes picked by hand? Where stems collected…please provide more information.
Response: Thanks for your issue. Healthy and undamaged grapes of 9 varieties of Chardonnay (Cha), Riesling (Rie), Pinot nior (Pin), Gem (Gem), Cabernet Sauvignon (Cab), Zinfandel (Zin), Longan (Lon), Syrah (Syr), Merlot (Mer) were collected using sterile scissors at veraison. In this period, grapes begin to color and enlarge, corresponding to stage 35 in the modified E-L system for identifying major and intermediate grapevine growth stages[29]. Considering the heterogeneity of the tested grapes, the grape samples were collected from at least 5 plants to form a composite sample, and five composite samples of grapes were collected for each variety of grape, all the samples were placed in sterile plastic bags, transported to the laboratory in refrigerated boxes and processed within 12 h.
Point 13: Lines 325–327 “The physicial chemical characteristics of the grape wine made from 7 varieties of the grapes collected in this study were as follows:”
The table should be cited in the text and not included in the text. Also, it is “physico-chemical”, but I do not think that is how this should be phrased because it is more of the composition of the grapes. I would just cite the table at the end of the first sentence in this paragraph. Example “Healthy and undamaged grapes of 9 varieties of Chardonnay(Cha), Riesling(Rie), Pinot nior(Pin), Gem(Gem), Cabernet Sauvignon(Cab), Zinfandel(Zin), Longan (Lon), Syrah (Syr), Merlot(Mer) were collected at veraison.”
I also have three concerns with the table.
Why are Chardonnay and Merlot types left out? Why is Pinot nior and Zinfandel missing data? I would present the wines types in a consistent order throughout the paper.
Response: Thanks for you issues. We could not obtain the integrated information of the wines from winery. Only the physico-chemical of the 7 varieties were obtained by the winery. Due to the samples in this study were the grow season at verasion, not harvest season, the table was removed in the manuscript.
Point 14: Lines 360–361 “Diversity between samples (beta diversity) was estimated by a series of tests, including ANOVA and Tukey’s test, which were performed using Bray-Curtis distances.”
Bray-Curtis distances does not exist. We use a Bray-Curtis dissimilarity metric and show distance with an ordination plot.
Response: Thanks for your issue. The express “Bray-Curtis distances” have been changed to “We use a Bray-Curtis dissimilarity metric and show distance with an ordination plot”.
Conclusion
Point 15: Lines 371–373 “The bacteria mainly distributed in eight phyla, including Proteobacteria, Actinobacteria, Firmicutes, and six genera, such as Erwinia, Pseudomonas, Bacillus Cohn, and so on.”
Please rephrase to include the names of all phyla and genera. Otherwise, it places emphasis on these taxa.
Response: Thanks for your issue. The sentence has been changed to “The dominant phyla of bacteria were Proteobacteria, Actinobacteria, Firmicutes, Gemmatimonadete and Bacteroidetes and the genera were Pseudomonas, Arthrobacter, Bacillus, Pantoea, Planomicrobium, Massilia, Curtobacterium, Corynebacterium, Cellulomonas, Sphingomonas and Microvirga.”
Point 16: The abstract and conclusions do not match up:
Abstract - “The dominated phylum of different wine grape carposphere were Proteobacteria, Actinomycetes, Firmicutes, Gemmatimonadete and Bacteroidetes. The dominant bacterial genera of different wine grape varieties were Pseudomonas, Arthrobacter, Bacillus, Pantoea, Planomicrobium, Massilia, Curtobacterium, Corynebacterium, Cellulomonas, Sphingomonas and Microvirga. The fungal communities of the grapes were dominated by Ascomycota, Basidiomycota, Zygomycota for all nine wine varieties. The dominant fungal genera on grape carposphere were Alternaria, Cladosporium, unclassified Capnodiales, Phoma, Rhodotorula, Cryptococcus, Aureobasidium and Epicoccum”
Conclusion – “The bacteria mainly distributed in fine phyla, including Proteobacteria, Actinobacteria, Firmicutes, Gemmatimonadete and Bacteroidetes and eleven genera, such as Erwinia, Pseudomonas, Bacillus Cohn, and so on. The dominant phyla of the fungi were Ascomycota, Basidiomycota, Zygomycota, and the genera were Alternaria, Cladosporium, unclassified Capnodiales, Phoma, Rhodotorula, Cryptococcus, Aureobasidium and Epicoccum. The most similar species of the bacterial core OTUs were Arthrobacter sp., Pseudomonas sp., and Cladosporium sp., Phoma sp., Alternaria sp”
Response: Thanks for your issue. We have checked with the abstract and conclusion carefully, and modified the conclusion with: “The bacteria mainly distributed in fine phyla, including Proteobacteria, Actinobacteria, Firmicutes, Gemmatimonadete and Bacteroidetes and eleven genera, such as Erwinia, Pseudomonas, Bacillus Cohn, and so on. The dominant phyla of the fungi were Ascomycota, Basidiomycota, Zygomycota, and the genera were Alternaria, Cladosporium, unclassified Capnodiales, Phoma, Rhodotorula, Cryptococcus, Aureobasidium and Epicoccum. The most similar species of the bacterial core OTUs were Arthrobacter sp., Pseudomonas sp., and Cladosporium sp., Phoma sp., Alternaria sp., and the most similar species of the fungal core OTUs were Alternaria sp. and Cladosporium sp.”
Point 17: Lines 367–368 “From the results, it is inferred that the microorganisms existed on the carposphere of the wine grapes may be beneficial or harmful on the health of the grape vine, the quality of the grape berries and the process of wine brewing.”
Analysis of the taxonomy is good, but where is the interpretation? This conclusion is not helpful and based on the relative abundances of the top taxa you could easily make assumptions as to their impact.
Response: Thanks for your issue. The genera of Alternaria was more enriched in the wine grape we studied though difference in relative abundance. Alternaria produces alternariol, alternariol monomethyl, altenuene, altertoxin, and tenuazonic acid, these metabolites exhibit some degree of toxicity to mammalian and bacterial cells as well as to higher plants, while Fusarium sp. are involved in wine making and can produce pectinase, raising juice yields during the process of wine making. In our study, all of these two potential functional microorganisms were existed in nine varieties of wine grapes. We have added the sentence “Alternaria produces alternariol, alternariol monomethyl, altenuene, altertoxin, and tenuazonic acid, these metabolites exhibit some degree of toxicity to mammalian and bacterial cells as well as to higher plants [12]. Fusarium sp. are involved in wine making and can produce pectinase, raising juice yields during the process of wine making [13,14]. In our study, all of these two potential functional microorganisms were existed in nine varieties of wine grapes.” In section 3.1. and the references added here: 12. Zarraonaindia, I., Owens, S.M., Weisenhorn, P., West, K., Hampton-Marcell, J., Lax, S., Bokulich, N.A., Mills, D.A., Martin, G., Taghavi, S., et al. The soil microbiome influences grapevine-associated microbiota. Mbio. 2015; 6(2): e02527-14, doi: 10.1128/mBio.02527-14. 13. Motta, S.D., and Valente Soares, L. M. Survey of Brazilian tomato products for alternariol, alternariol monomethyl ether, tenuazonic acid and cyclopiazonic acid. Food Addition tam B. 2001; 8, 630-634. doi:10.1080/02652030117707. 14. Fredj, S.M.B., Chebil, S., Lebrihi, A., Lasram, S., Ghorbel, A., and Mliki, A. Occurrence of pathogenic fungal species in Tunisian vineyards. Int J Food Microbiol. 2007; 113, 245-250. doi: 10.1016/j.ijfoodmicro.2006.07.022.

Reviewer 2 Report
Few typing errors should be corrected. Passive voice should be also used eg. "we used LEfSe to indentify ...." should be "LEfSe was used to .."
Author Response
Few typing errors should be corrected. Passive voice should be also used eg. "we used LEfSe to indentify ...." should be "LEfSe was used to .."
Response: Thanks for your advise. We have changed the sentences “we used LEfSe to indentify” to "LEfSe was used to .."

Round 2
Reviewer 1 Report
I am very happy with the revisions the authors have made. I appreciate that they addressed my comments and the figures look better. The publication reads better, but there are still some awkward sentences and some grammar mistakes that need attention. I do still have one concern and it is in regards to “point 1” in their revision comments.
“Point 1: Please state that the microbes were surveyed with 16S rRNA gene and ITS analyses.
Response: Thanks for your issue. We have added the sentence “In this work, high-throughput sequencing analysis was performed to investigate the differences of epiphytic microbial communities inhabiting different varieties of wine grape berries” in the Abstract.”
I had previously requested for mention of methods (16S rRNA and ITS) for transparency. Inclusion of “high-throughput sequencing analysis” is not sufficient because they are many methods including metagenomes that use high-throughput sequencing. I specifically want this stated, because I have seen many researchers get confused over this and without it explicated sated it might get cited in the future incorrectly.
Author Response
Thanks for your issue. We have changed the sentence to “In this work, 16S rRNA and ITS analysis were performed to investigate the differences of epiphytic microbial communities inhabiting different varieties of wine grape berries” in the Abstract.
